# Genetic modulation of RNA splicing rescues BRCA2 function in mutant cells

Beatriz Anjo Lima[1], Ana Carolina Pais[1], Juliette Dupont[3] , Patrícia Dias[3], Noélia Custódio[1,2] , Ana Berta Sousa[3] , Maria Carmo-Fonseca[1,2] , Célia Carvalho[1,2] 

Variants in the hereditary cancer-associated *BRCA1* and *BRCA2* genes can alter RNA splicing, producing transcripts that encode internally truncated yet potentially functional proteins. However, few studies have quantitatively analyzed variant-specific splicing isoforms. Here, we investigated cells heterozygous and homozygous for the *BRCA2*:c.681+5G>C variant. Using droplet digital RT–PCR, we identified two variant-specific mRNA isoforms. The predominant transcript is out-of-frame, contains a premature termination codon, and is degraded via the nonsense-mediated mRNA decay pathway. In addition, we detected a novel minor isoform encoding an internally truncated protein lacking non-essential domains. Homozygous mutant cells expressed low levels of BRCA2 protein and were defective in DNA repair. Using CRISPR-Cas9 gene editing, we induced the production of in-frame transcripts in mutant cells, which resulted in increased protein expression, enhanced RAD51 focus formation, and reduced chromosomal breaks after exposure to genotoxic agents. Our findings highlight the therapeutic potential of splicing modulation to restore BRCA2 function in mutant cells, offering a promising strategy to prevent cancer development.

## Introduction

Germline mutations in breast cancer genes *BRCA1* and *BRCA2* significantly increase the likelihood of developing particular types of malignancies, namely, breast and ovarian cancer (Chen & Parmigiani, 2007; Stratton & Rahman, 2008). Both the proteins encoded by the *BRCA1* and *BRCA2* genes are required for the repair of DNA double-stranded breaks (DSBs) by homologous recombination (Prakash et al, 2015). In order to restore the correct sequence, DSBs are first end-resected to generate 3′ single-stranded DNA. The strand-exchange protein RAD51 then binds the single-stranded DNA forming a nucleoprotein filament that invades an intact homologous DNA duplex, typically the identical sister chromatid (Gorodetska et al, 2019). In the absence of functional BRCA1 or BRCA2, DSBs tend to be repaired by non-homologous end joining (NHEJ) in which the two broken DNA ends are joined without using a homologous DNA sequence to guide repair. Compared with homologous recombination repair (HRR), NHEJ is much more error-prone and often results in the introduction of DNA mutations, namely, DNA deletions or insertions, which can affect the function of critical cancer genes thereby driving oncogenesis (Gorodetska et al, 2019).

Most deleterious mutations that have been identified in the *BRCA1* and *BRCA2* genes introduce premature termination codons (PTCs) through small frameshift deletions or insertions, nonsense or splice junction alterations, or large deletions or duplications (Stratton & Rahman, 2008). PTCs may result in the production of carboxy-terminally truncated proteins that compromise their function. However, most transcripts that harbor nonsense codons are eliminated by a surveillance mechanism termed nonsense-mediated mRNA decay (NMD). NMD is a translation-dependent process that recognizes the presence of a PTC and triggers the degradation of the mRNA (Maquat, 2004). In the nucleus, shortly after splicing of pre-mRNAs, each exon–exon junction site is coated with a protein complex called the exon junction complex (EJC; Le Hir et al, 2000). In the cytoplasm, newly synthesized mRNAs undergo a pioneer round of translation during which ribosomes displace EJCs. Given that most of the normal stop codons reside in the last exon of a gene, the presence of an EJC downstream of a terminating ribosome acts as a trigger for NMD (Kurosaki & Maquat, 2016). Generally, PTCs that are located within mRNA at a position that is more than 50–55 nucleotides upstream of an exon–exon junction elicit the exonucleolytic degradation of the mRNA (Isken & Maquat, 2007).

Although germline variants that introduce PTCs in transcripts from high-risk cancer susceptibility genes tend to be considered as deleterious because of predicted loss of function of the mutant allele, there are exceptions to the rule; namely, several lines of evidence indicate that naturally occurring and variant-related alternative splicing events in *BRCA1* and *BRCA2* transcripts can encode protein isoforms with residual activity (Mesman et al, 2020).

[1]Faculdade de Medicina da Universidade de Lisboa, Lisboa, Portugal [2]GIMM - Gulbenkian Institute for Molecular Medicine, Lisbon, Portugal [3]Serviço de Genética, Unidade Local de Saúde Santa Maria, Centro Académico de Medicina de Lisboa, Lisboa, Portugal

Correspondence: celiacarv@edu.ulisboa.pt; carmo.fonseca@medicina.ulisboa.pt

Moreover, variant alleles may be associated with production of multiple splicing isoforms. However, a systematic quantification of variant-associated mRNA isoform ratios combined with measurements of protein functional activity has not been performed. Here, we used the *BRCA2*:c.681+5G>C variant as an experimental model system for two main reasons. First, this variant was previously shown to affect splicing (Quiles et al, 2016), but it is still classified as a variant of unknown significance in the ClinVar database (Landrum et al, 2016) (ClinVar, https://www.ncbi.nlm.nih.gov/clinvar/variation/483115/, last accession on december 27, 2024). Second, we had access to cells from individuals with clinical features suggestive of a hereditary cancer syndrome who are either hetero- or homozygous carriers of the variant.

To quantify variant-associated splicing isoforms, we used droplet digital RT–PCR (ddRT–PCR), a technology that provides orders of magnitude more precision and sensitivity than real-time PCR (qPCR) after retrotranscription (RT) (Hindson et al, 2011). The high sensitivity of ddPCR technology is particularly useful for improving the detection of rare nucleic acid molecules. Moreover, error rates in ddPCR are significantly reduced and tolerance to amplification inhibitors is increased, when compared to qPCR methods (Taylor et al, 2017). Partition of RNA samples into thousands of droplets, each containing all the elements necessary for retrotranscription and PCR amplification, evades competition between splicing isoforms and favors unbiased amplification of under-represented molecules.

To assess the function of the BRCA2 protein in mutant cells, we quantified the formation of RAD51 nuclear foci in response to DNA damage. When cells are exposed to ionizing radiation or to the topoisomerase II inhibitor etoposide, they accumulate DSBs in DNA (Olive & Banáth, 1993). The efficient repair of these lesions involves the recruitment of RAD51 to the sites of damaged DNA, forming distinct foci that can be visualized by microscopy (Tarsounas et al, 2004). The formation of RAD51 foci in response to DNA damage is dependent upon the function of BRCA2 in homologous recombination (Yuan et al, 1999; Tarsounas et al, 2003; Ceccaldi et al, 2016). Thus, the inability to form RAD51 foci can be used as a marker of BRCA2 deficiency (Davies et al, 2001; Godthelp et al, 2006).

In addition, we quantified chromosome abnormalities in metaphase spreads from cells treated with mitomycin C, a drug that induces interstrand crosslinks in the DNA. In human cells, these DNA lesions are normally repaired by the Fanconi anemia (FA) pathway (Wang, 2007). Several genes are involved in the FA pathway, including *BRCA2* (Niraj et al, 2019), and cells from patients with biallelic *BRCA2* deleterious mutations are hypersensitive to mitomycin C (MMC) and other DNA cross-linking agents such as diepoxybutane (DEB) or cisplatin (Howlett et al, 2002; Oostra et al, 2012).

Our results reveal that the *BRCA2*:c.681+5G>C variant abolishes the expression of the canonical mRNA and induces the expression of at least two variant-specific splicing isoforms. A major isoform generates a PTC and targets the transcript for NMD, whereas a novel minor isoform is predicted to encode an internally truncated functional protein. Homozygous mutant cells expressed low levels of BRCA2 protein and were defective in DNA repair, as shown by reduced formation of RAD51 foci in response to etoposide and increased chromosomal abnormalities induced by mitomycin C. This suggests that the protein encoded by the minor splicing

isoform is not sufficient to sustain DNA repair activity. Using CRISPR-Cas9 genome editing to induce an additional exon skipping in cells carrying the biallelic variant, we show that increasing the expression of in-frame transcripts resulted in improved DNA repair function. Thus, strategies designed to induce the expression of in-frame transcripts that lack non-essential exons have the potential to rescue BRCA2 function in mutant cells, setting the basis for preventive interventions in individuals at high risk for hereditary cancer.

# Results

### Identification of a novel splicing isoform associated with the *BRCA2*:c.681+5G>C variant

A 66-yr-old female (LP) was referred to the Genetics Clinic at Hospital de Santa Maria because of the history of bilateral breast cancer. She was first diagnosed with cancer in the right breast at the age of 45 and later in the left breast at 57 yr. Family history was unremarkable, without cases of breast, ovarian, prostate, or pancreatic cancer known. Her father died at the age of 68 because of colon cancer, and one paternal uncle had a malignant thyroid tumor (Fig 1A). LP's *BRCA1* and *BRCA2* genes were screened by next-generation sequencing and multiplex ligation-dependent probe amplification. No pathogenic or likely pathogenic variants were identified, but the variant of unknown significance *BRCA2*:c.681+5G>C (NM_000059.4) was detected in heterozygosity. This variant results from a G-to-C substitution at nucleotide +5 in intron 8 (Fig 1B). The splicing prediction tool SpliceAI (Jaganathan et al, 2019) predicts that this variant significantly weakens both the upstream donor and acceptor natural splice sites. A previous study analyzed the *BRCA2* mRNA expressed in PBMCs from an independent person carrying this variant and found skipping of exon 8, generating an out-of-frame deletion and creation of a premature stop codon in exon 9 (Quiles et al, 2016).

To quantify the relative proportion of *BRCA2* mRNA splicing isoforms expressed in LP PBMCs, we used ddRT–PCR. We designed exon boundary probes and primers spanning the junctions between exons 7 and 8 (probe e7e8), exons 7 and 9 (probe Δ8), and exons 13 and 14 (probe e13e14) of *BRCA2* transcripts. In control *BRCA2* WT cells, the number of mRNA molecules including exons 7 and 8 was similar to the number of molecules including exons 13 and 14, indicating that all of mRNAs that include exons 13 and 14 also include exons 7 and 8 (Fig 2A). This proportion was not significantly altered by treatment with the translation inhibitor cycloheximide (Fig 2A), as expected for mRNAs that are not targets of NMD. In LP cells, only ~50% of mRNAs detected by the e13e14 probe included exons 7 and 8, and the proportion was not significantly altered by cycloheximide treatment (Fig 2A). These molecules most likely correspond to transcripts from the WT *BRCA2* allele. The probe targeting the junction between exons 7 and 9 (probe Δ8) did not detect any transcripts in control cells. In contrast, in LP cells, ~15% of the mRNAs detected by the e13e14 probe were also detected by the Δ8 probe (Fig 2B). After cycloheximide treatment, this proportion increased to ~29% (Fig 2B), consistent with the

**A**

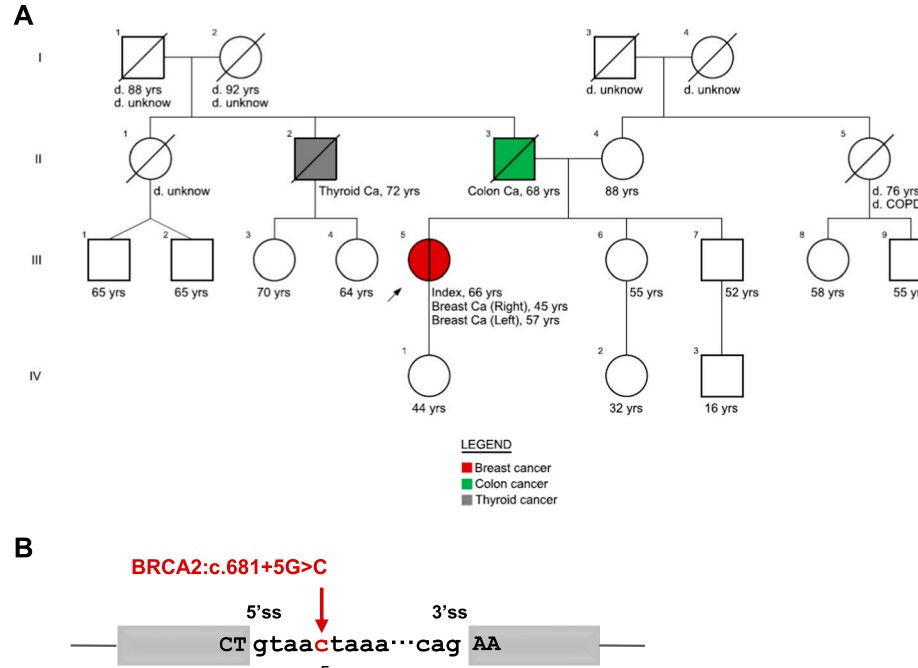

**B**

BRCA2:c.681+5G>C

5'ss        3'ss

CT gtaa**c**taaa···cag AA

Exon 8    +5    Exon 9

**Figure 1. *BRCA2*:c.681+5G>C variant.**
**(A)** Pedigree of LP's family. The proband is indicated with an arrow. Circles represent female family members, and squares represent male family members; a slash indicates deceased individuals. Colors denote cancer diagnoses, with the age at death (for those deceased) and age at cancer diagnosis specified. The current age of the proband is also provided. **(B)** Schematic representation of the variant's location within intron 8 of the *BRCA2* gene. Donor (5') and acceptor (3') splice sites (ss) are indicated.

LEGEND
■ Breast cancer
■ Colon cancer
■ Thyroid cancer

presence of a PTC that is predicted to activate NMD (Fackenthal et al, 2016). However, we were intrigued by the low level of mutant mRNA detected by the Δ8 probe, even after cycloheximide treatment. Because there is a naturally occurring alternative mRNA isoform termed *BRCA2* Delta6q7 (r.478_631del154) that lacks most of exon 6 (except for two nucleotides, TG) and skips exon 7 completely (Fackenthal et al, 2016), we hypothesized that this alternative splicing pattern could also occur in mutant transcripts in combination with exon 8 skipping. Thus, we designed forward and reverse primers on exons 5 and 10, respectively, to amplify cDNA from LP PBMCs treated with cycloheximide. Sanger sequencing of fragments separated by gel electrophoresis revealed the canonical splicing isoform and two additional aberrant transcripts, one with complete skipping of exon 8 (Δ8, r.632_681del50) (Fig 2C), and another that splices the first two nucleotides of exon 6 to exon 9 (Δ6q-8, r.478_681del204) (Fig 2D). This newly identified variant-specific isoform maintains the reading frame and potentially codes for an internally truncated protein (p.160-227del).

To quantify the relative abundance of these splicing isoforms, we designed additional exon boundary probes and primers spanning the junctions between exons 5/6p and 8 (probe Δ6q,7), and exons 5/6p and 9 (probe Δ6q-8). We found that in control cells, the *BRCA2* Delta6q,7 isoform corresponds to ~6% of all *BRCA2* transcripts (Fig 2E). After cycloheximide treatment, the proportion increases to ~14%, consistent with the presence of a PTC that fulfills the requirements to be degraded by NMD (Fackenthal et al, 2016). In LP cells, this isoform corresponds to ~3% of all *BRCA2* transcripts and increases to ~6% after cycloheximide treatment (Fig 2E). This suggests that *BRCA2* Delta6q,7 mRNAs detected in LP cells are transcribed from the WT allele only. The probe Δ6q-8 does not detect any mRNA in control cells, but in LP cells, it reveals an

isoform representing 8–10% of all *BRCA2* transcripts detected by probe e13e14 (Fig 2F). We conclude that out-of-frame splicing of exons 7 and 9 is not the only splicing change associated with the *BRCA2*:c.681+5G>C variant. At least one additional splicing isoform was identified, which skips most of exon 6 and exons 7 and 8. Delta6q-8 mRNAs were not detected in control cells, suggesting that they are transcribed from the mutant allele. Notably, this splicing pattern does not perturb the open reading frame. Therefore, the resulting mRNA has the potential to encode an internally shortened but still functional BRCA2 protein.

## Cells homozygous for *BRCA2*:c.681+5G>C are deficient in DNA repair

Having shown that ~10% of mRNAs transcribed from the mutant allele in heterozygous cells encode a potentially functional protein, we asked whether this small proportion of transcripts is sufficient to maintain HRR activity. To address this question, we analyzed cells from two sisters (here referred to as CC and DC) who are homozygous for the *BRCA2*:c.681+5G>C variant. The parents of these sisters were first-degree cousins (Fig 3A). Their father, who was a heavy smoker, died at 58 yr of age of lung cancer, a paternal female cousin died at 38 yr of age with breast cancer, a maternal aunt was diagnosed with breast cancer at 55 yr of age, maternal grandmother died of gastric cancer at 55 yr of age, and the maternal grandfather died at 66 yr of liver cancer diagnosed at 64 yr. Their mother had no history of cancer at 70 yr of age. One of the sisters (CC) was diagnosed at 29 yr with right breast cancer. No pathogenic or likely pathogenic variants in the *BRCA1* and *BRCA2* genes were identified in addition to the homozygous *BRCA2*: c.681+5G>C variant of uncertain significance. At examination, this

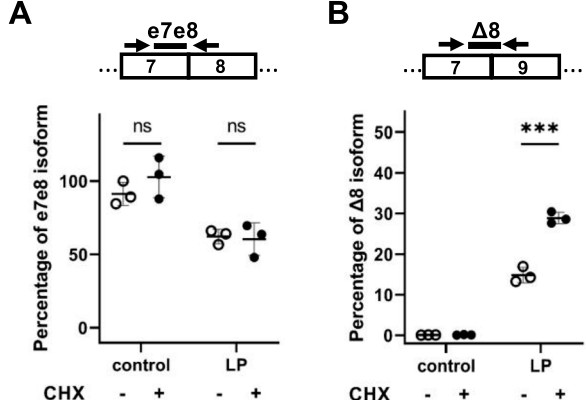

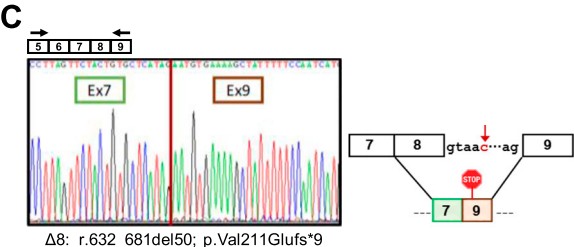

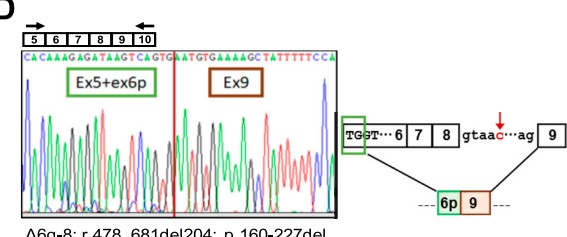

Δ8: r.632_681del50; p.Val211Glufs*9

Δ6q-8: r.478_681del204; p.I60-227del

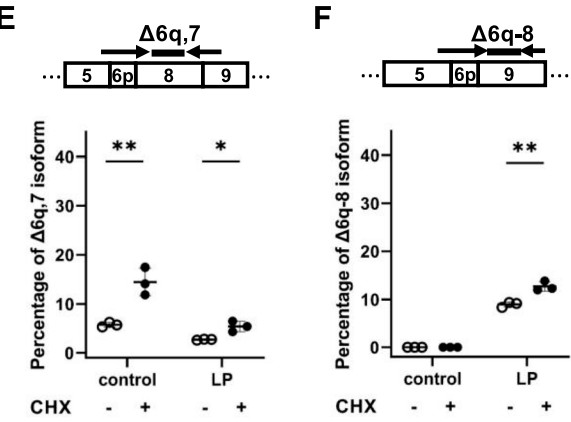

**Figure 2. Splicing isoforms expressed in PBMCs from a heterozygous carrier of the *BRCA2*:c.681+5G>C variant.**
**(A, B, E, F)** PBMCs from a control individual and from LP were either mock-treated or treated with cycloheximide (CHX). Total RNA was isolated and analyzed by ddRT–PCR. Probes and primers are indicated. Graphs show the percentage of the indicated isoform relative to total *BRCA2* transcripts detected by a probe spanning the junction between exons 13 and 14. Error bars represent the SD. Significant differences: *$P < 0.05$; **$P < 0.01$; ***$P < 0.001$; ns, non-significant. **(C, D)** Sanger sequencing of the two aberrant *BRCA2* cDNA fragments isolated from

patient had no physical abnormalities associated with Fanconi anemia and hematological and endocrinological evaluations were normal. Cytogenetic testing of lymphocytes with diepoxybutane (DEB) revealed an isolated radial figure and an increased number of breaks per cell below the reference values for a positive diagnosis of Fanconi anemia. The patient was also tested with the TruSight Cancer Panel (Illumina) targeting 94 genes and 284 SNPs associated with a predisposition toward various cancers, and no pathogenic variant was identified. The other sister (DC) was diagnosed at 39 yr with sigmoid adenocarcinoma. Tumor testing for mismatch repair proteins by immunohistochemistry was normal. Cytogenetic testing of lymphocytes with DEB did not show any radial figures, and similar to her sister, an increased number of breaks per cell were observed. At the last appointment, this patient had performed prophylactic bilateral salpingo-oophorectomy, had no history of breast cancer, and was diagnosed with 18-mm non-functional adrenal adenoma. This patient also did not exhibit any physical abnormalities characteristic of Fanconi anemia. Segregation analysis of the variant in family members could not be performed.

Contrasting to heterozygous LP cells (Fig 2), quantification of splicing isoforms expressed in PBMCs from CC and DC revealed residual levels (<2%) of mRNA including exon 8 (Fig 3B), indicating that the mutant allele is unable to produce WT mRNA. The probe Δ6q,7 also failed to detect any transcripts (Fig 3B), showing that the minor Delta6q7 isoform produced from the WT allele cannot be generated from the mutant allele. Rather, the variant induces production of two aberrant splicing isoforms, one that lacks exon 8, detected by the probe Δ8, and another that splices the first two nucleotides of exons 6 to 9, detected by the probe Δ6q-8 (Fig 3B). Determining whether the 20% of in-frame Delta6q-8 mRNAs expressed in homozygous cells encode a potentially functional protein required extended analysis.

To have an unlimited source of cells and perform functional assays, we derived lymphoblastoid cell lines (LCLs) from the two sisters. As controls, we used LCLs harboring WT *BRCA2* alleles. Isoform quantification by ddRT-PCR confirmed that mRNAs containing exon 8 were not expressed in CC and DC cells (Figs 3C and S1). In both CC and DC cells, the probe Δ8 detected ~30% of mRNAs, and after cycloheximide treatment, the proportion increased to ~60% (Figs 3C and S1). When we restricted the analysis to nuclear RNAs, ~50% of mRNAs hybridized to the Δ8 probe, and this proportion did not significantly increase after cycloheximide treatment (Fig 3D). This is in agreement with several lines of evidence indicating that degradation of mRNAs targeted by NMD occurs in the cytoplasm (Kurosaki & Maquat, 2016). As observed in PBMCs (Fig 3B), CC-derived lymphoblastoid cells do not express the *BRCA2* Delta6q7 isoform (Figs 3C and D and S1). Rather, ~20% of transcripts lack most of exon 6 and exons 7 and 8 (Figs 3C and D and S1). The relative abundance of this isoform is not significantly altered by treatment with cycloheximide (Figs 3C and S1), as expected for mRNAs that are not targets of NMD.

LP PBMCs. The mRNA positions (Human Genome Variation Society nomenclature) and protein consequences are indicated. Schematic representation of the detected isoforms.
Source data are available for this figure.

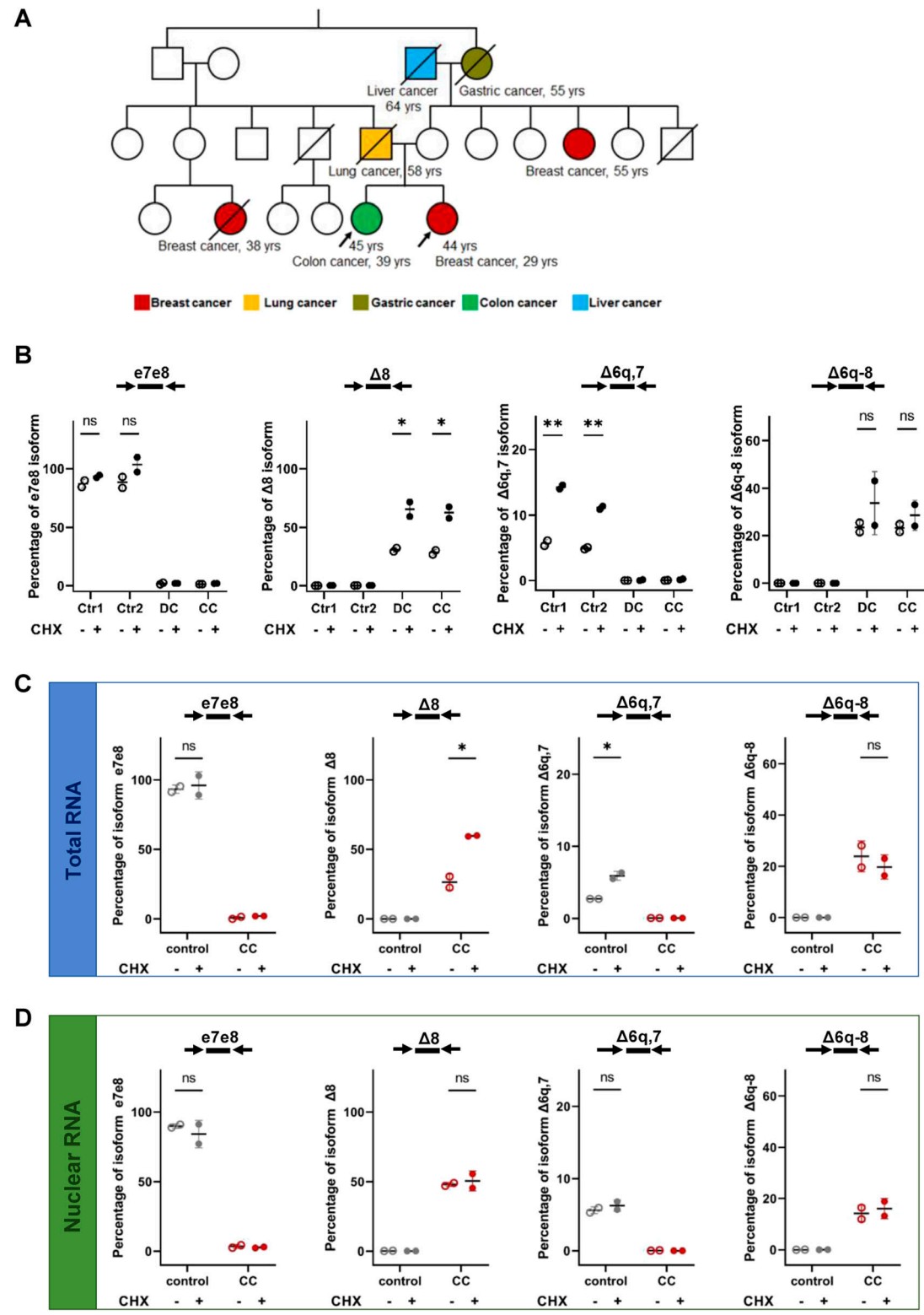

**Figure 3. Splicing isoforms expressed in cells from homozygous carriers of the *BRCA2*:c.681+5G>C variant.**
**(A)** Pedigree of CC and DC's family, two sisters who are homozygous carriers of the *BRCA2*:c.681+5G>C variant. The probands are indicated with arrows. Circles represent female family members, and squares represent male family members; a slash indicates deceased individuals. Colors denote cancer diagnoses, with the age at cancer diagnosis specified. The current age of the probands is also provided. **(B, C, D)** Quantification by ddRT–PCR of splicing isoforms expressed in PBMCs (B) and lymphoblastoid cell lines (C, D) from control individuals and DC and CC patients. Probes and primers are indicated. Cells were either mock-treated or treated with cycloheximide (CHX). Graphs on panel B show the percentage of the indicated isoform relative to total *BRCA2* transcripts detected by a probe spanning the junction between exons 13 and 14.

Consistent with the prediction that these mRNAs can be translated into an internally truncated potentially functional protein, immunoblot analysis revealed the presence of BRCA2 protein in both control and CC-derived lymphoblastoid cells (Fig 4A). BRCA2 expression is regulated during the cell cycle and is proportional to the rate of cell proliferation (Bertwistle et al, 1997; Su et al, 1998). Thus, we compared the levels of BRCA2 protein with cyclin A, which is up-regulated during the S phase (Desdouets et al, 1995). We observed BRCA2 bands of similar molecular weight in both control and mutant cells (Fig 4A). However, BRCA2 is a 384-kD protein composed of 3,418 amino acids, of which only 68 are encoded by exons 6, 7, and 8. Therefore, the small difference in size between the full-length and internally truncated proteins is not sufficient to distinguish them on immunoblots.

To assess whether the BRCA2 protein expressed in CC-derived lymphoblastoid cells is sufficient to maintain DNA repair activity, we quantified RAD51 nuclear foci formed after induction of DNA DSBs by etoposide. Cells were treated with 5 $\mu$M etoposide for 15 min and analyzed 6 h later to allow the assembly of repair complexes. DSBs were visualized by immunofluorescence using antibodies to histone H2AX phosphorylated at Ser 139 ($\gamma$H2AX). At 30 min after the 15-minute exposure to etoposide, the proportion of cells that exhibited multiple discrete $\gamma$H2AX nuclear foci increased sharply (Fig 4B). At 6 h after the 15-minute exposure to etoposide, double labeling showed co-localization of RAD51 and $\gamma$H2AX foci. Approximately 20% of control cells had 10 or more RAD51 nuclear foci, contrasting with less than 1% of patient cells (Fig 4C). Notably, very low numbers of $\gamma$H2AX foci were observed at 24 h after etoposide treatment in both control and patient-derived cells (Fig S2), indicating that most of the DNA lesions were repaired. The reduced ability of CC- and DC-derived cells to form RAD51 foci 6 h after exposure to etoposide suggests a specific defect in HRR.

In conclusion, we show that ~60% of BRCA2 mRNAs expressed in homozygous mutant cells contain exon 7 spliced to exon 9 generating a premature stop codon (Fig 3B, $\Delta$8), whereas ~20% of transcripts encode an internally truncated yet potentially functional protein (Fig 3B, $\Delta$6q-8). However, these mRNAs are not sufficient to maintain normal BRCA2 function in DNA repair, as assessed by recruitment of RAD51 to sites of DNA lesions. This prompted us to ask whether forcing the expression of alternative splicing isoforms that encode internally truncated proteins lacking non-essential domains could rescue the DNA repair function of BRCA2.

### Engineered exclusion of exon 7 in homozygous mutant cells partially restores DNA repair

Upon analyzing the BRCA2 gene sequence, we noted that the simultaneous skipping of exons 7 and 8 preserves the reading frame. Based on previous evidence that exons 7 and 8 do not encode essential parts of the BRCA2 protein (Mesman et al, 2020), forcing the exclusion of exon 7 in cells that skip exon 8 because they have

the BRCA2:c.681+5G>C variant could, in principle, restore the protein function. To address this hypothesis, we used the CRISPR-Cas9 technology to disrupt the splice sites of exon 7 (Fig 5A). We identified four suitable PAM sequences targeted by four different guide RNAs that would direct DSBs directly adjacent to either exon 7 acceptor or donor splice sites (Fig 5A and see the Materials and Methods section). Without a template for repair, the error-prone NHEJ pathway is expected to be preferentially used, introducing mutations that disrupt these splice sites. We designed guide RNAs consisting of a fluorescently labeled tracrRNA with a constant sequence and a crRNA with a sequence specific for the target site. Patient-derived cells were transfected via electroporation with ribonucleoproteins comprising each guide RNA and the Cas9 nuclease. The transfection efficiency was assessed by flow cytometry 1 d after electroporation. 1 wk after transfection, a sample of cells was harvested for RNA analysis and the remainder were frozen for future characterization. Quantification by ddRT–PCR revealed that compared with the parental non-edited CC cells, gene-edited cells expressed lower levels of mRNA containing exon 7 (Fig S3A), suggesting efficient destruction of the targeted splice sites. In contrast, as expected, edited cells expressed higher levels of isoform Delta7,8 (Fig 5B). The proportion of BRCA2 transcripts having exon 6 spliced to exon 9 ranged between 20% and 50%, depending on the guide RNA (Fig 5B). This suggests that the gene editing intervention induced additional alternative splice patterns. From the polyclonal population of cells edited with gRNA7, monoclonal populations were selected by serial dilution. Isoform quantification by ddRT–PCR confirmed that Delta7,8 was the new predominant splicing isoform in at least seven cell clones (Clones 1, 4, 5, 9, 12, 15, and 16; Fig S3B). These clones, which also exhibited the minimal expression of the isoforms $\Delta$8 and $\Delta$6q-8 (Fig S3B), were selected for further characterization. Sequencing of cDNA confirmed the expression of the Delta7,8 isoform (r.517-681del165; p.173-227del) (Fig S3C), and increased BRCA2 protein expression was validated by immunoblot analysis (Fig S3D).

Next, we quantified RAD51 nuclear focus formation in gene-edited cells. Cells were treated with 10 $\mu$M etoposide for 15 min and analyzed 3 h later by confocal microscopy. Compared with the parental non-edited patient cells, edited clones 1, 12, and 15 showed a significantly higher proportion of cells with ≥10 RAD51 foci that co-localize with $\gamma$H2AX foci (Fig 5C and D). However, the observed proportions tend to be lower when compared to control cells, suggesting that the function of the BRCA2 protein in HRR was not fully restored by gene editing. Notably, the proportion of cells positive for RAD51 foci is consistently higher in this assay compared with Fig 4, where cells were treated with a lower concentration of etoposide (5 $\mu$M), indicating a dose-dependent increase in RAD51 focus formation. We in addition performed a chromosomal breakage test to compare the sensitivity of edited and non-edited patient cells with the chromosome-breaking effect of mitomycin C (Fig 5E–G). Typical chromosomal aberrations involve disruptions affecting either one or both sister chromatids, along with

---

**(C, D)** For lymphoblastoid cells, total (C) or nuclear (D) RNA was isolated and analyzed. Given that nuclei are enriched for unspliced pre-mRNA, on panels C and D, the percentage is relative to the sum of spliced and unspliced BRCA2 (18–19 exon junction and exon 18–intron 18 boundaries, respectively). Error bars represent the SD. Significant differences: *$P < 0.05$; **$P < 0.01$; ***$P < 0.001$; ns, non-significant.
Source data are available for this figure.

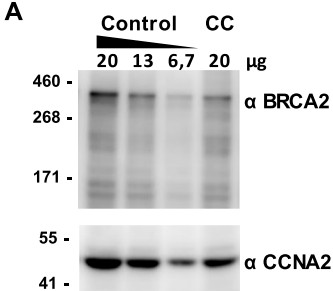

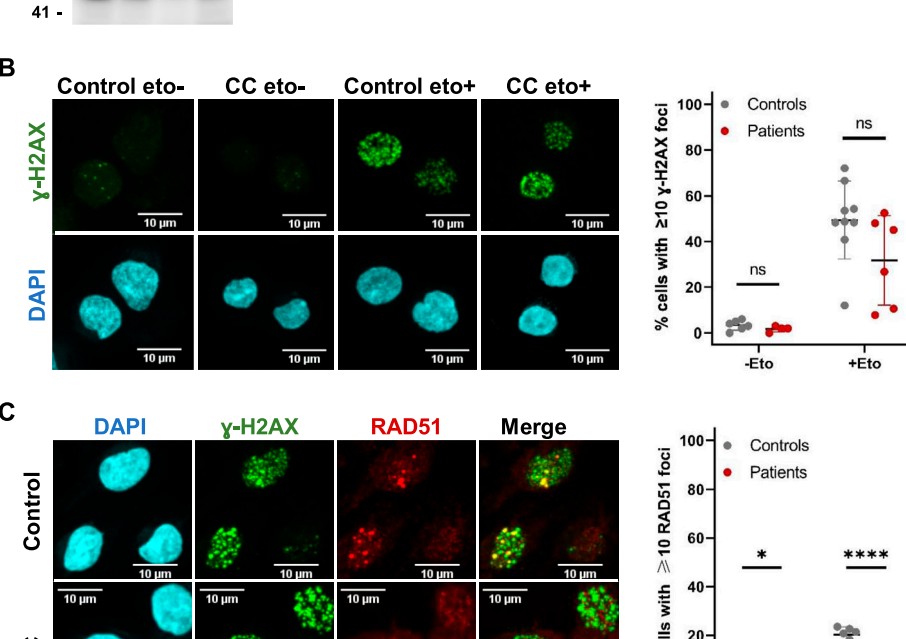

**Figure 4. Deficient recruitment of RAD51 to DNA lesions in homozygous cells.**
**(A)** Immunoblot detection of the BRCA2 protein in extracts from control and CC lymphoblastoid cell lines. Cyclin A was used as a loading control. Serial dilutions of the control cell line are shown. **(B, C)** Control and patient CC-derived lymphoblastoid cell lines were treated with 5 $\mu$M etoposide for 15 min. **(B)** At 30 min after etoposide treatment, cells were analyzed by immunofluorescence with anti-gamma-H2AX antibody; nuclei were stained with DAPI. The graph on the right quantifies cells with ≥10 nuclear $\gamma$-H2AX foci relative to the total number of cells. **(C)** At 6 h after etoposide treatment, cells were analyzed by immunofluorescence with anti-gamma-H2AX and anti-RAD51 antibodies; nuclei were stained with DAPI. The graph on the right quantifies cells with ≥10 nuclear RAD51 foci relative to the total number of cells. Dot plots in (B, C) depict results from multiple assays, including two to three replicates using three control cell lines, and CC and DC patient cell lines. Each dot represents an individual replicate. Over 200 cells were counted per assay. Error bars represent the SD. Significant differences: *$P < 0.05$; **$P < 0.01$; ***$P < 0.001$; ****$P < 0.0001$; ns, non-significant.
Source data are available for this figure.

formations that are either triradial, quadriradial, or more complex (Fig 5E). The number of chromosomal breaks and aberrations was quantified in each cell type. As shown in Fig 5F and G, we observed less abnormal events in metaphase spreads from edited cell clones compared with non-edited patient cells. This trend became even more evident upon calculating the percentage of aberrant cells and the average number of break events per cell, with several clones showing a pattern more similar to control cells than to non-edited patient cells (Fig 5F and G). Unexpectedly, clones 4 and 9, which do not restore the ability to form RAD51 foci (Fig 5D), showed reduced chromosome breakage in response to mitomycin C (Fig 5F and G). We consider several potential explanations for this observed discrepancy. One possibility is that the assays capture distinct aspects of BRCA2 functionality, with RAD51 focus formation directly reflecting HRR capacity, whereas the mitomycin C–induced chromosome breakage assay may involve additional pathways or compensatory mechanisms. Another possibility is threshold sensitivity, where subtle differences in BRCA2 activity might suffice to reduce chromosome breakage but not support detectable RAD51 focus formation. A third explanation involves clone-specific epigenetic or genetic variations that could differentially affect DNA repair pathways.

Although the precise reasons for the discrepancy remain unclear, our findings collectively indicate that increasing the expression of internally truncated protein isoforms in mutant cells can partially rescue BRCA2 function.

## Discussion

Germline nonsense variants, frameshift deletions or insertions, and single nucleotide variants located at or near splice sites in disease-causing genes can lead to loss of function because they generally introduce PTCs that target the mutant transcripts for degradation by NMD. However, in some cases, alternative splicing of pre-mRNAs transcribed from mutant alleles generates in-frame mRNAs that are translated to truncated functional proteins, as recently reported for *BRCA1* and *BRCA2* genes (De La Hoya et al, 2016; Seo et al, 2018; Mesman et al, 2020; Meulemans et al, 2020). In light of these findings, current interpretation guidelines for germline variants in

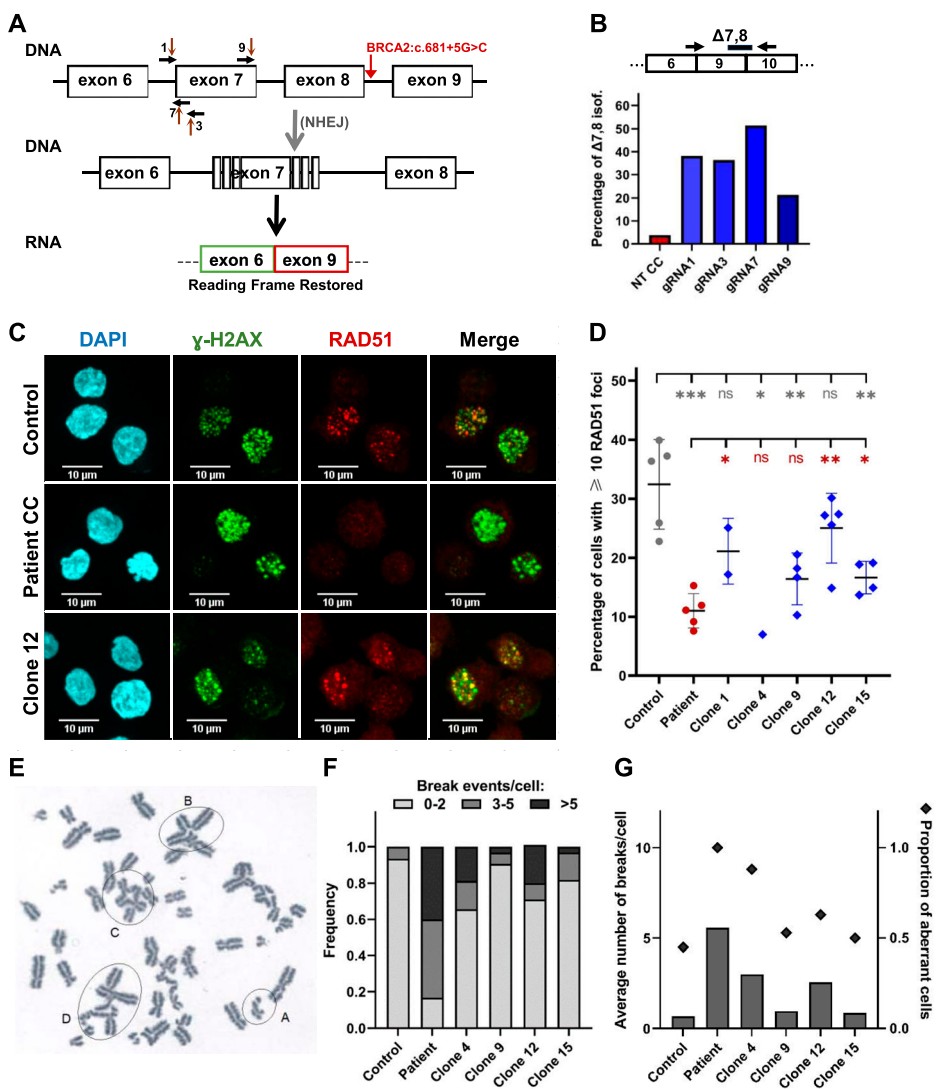

**Figure 5. Engineered exclusion of exon 7 in homozygous mutant cells partially restores DNA repair.**
**(A)** Schematic representation of the strategy used to disrupt exon 7 splice sites. **(B)** Quantification by ddRT–PCR of the Delta7,8 splicing isoform expressed in non-transfected CC cell line (red), and CC cell line transfected with the indicated guide RNAs (blue). Probe and primers are indicated. The graph shows the percentage of isoform detected by the indicated probe relative to total *BRCA2* transcripts detected by a probe spanning the junction between exons 13 and 14. **(C, D)** Indicated cells were treated with 10 μM etoposide for 15 min and analyzed 3 h later. Immunofluorescence with anti-gamma-H2AX and anti-RAD51 antibodies. Nuclei stained with DAPI. Confocal microscopy images after maximum intensity projection of Z-stacks of representative examples are shown. A control cell line was GM20172. The patient refers to CC-derived cell line. Edited clones were isolated from CC-derived cells transfected with gRNA 7. RAD51 foci were quantified in over 200 cells per assay. A dot plot shows results from multiple assays, with each dot representing an individual replicate. Significant differences: *P < 0.05; **P < 0.01; ***P < 0.001; ns, non-significant. **(E)** Metaphase spread illustrating chromosome alterations after treatment with mitomycin. (A, B, C, D) Circles highlight some of the more complex aberrations. **(F)** Quantification of chromosomal breakage events in the GM20172 control cell line, patient CC-derived cells, and edited clones 4, 9, 12, and 15. At least 20 metaphases were analyzed per cell type. **(G)** Mean number of chromosomal breaks per cell is shown as a bar graph with the y-axis on the left. The proportion of metaphases with aberrant events for each cell line/clone is shown as diamonds, with the y-axis on the right.
Source data are available for this figure.

*BRCA1* and *BRCA2* genes include alerts for potential in-frame rescue transcripts, namely, when the variants affect the splice site regions of *BRCA2* exons 4, 7, 8, 10, 12, and 14 (Mesman et al, 2020). However, no study to date has measured the abundance of in-frame rescue transcripts and the functionality of the encoded protein. Using the *BRCA2*:c.681+5G>C variant as a model system and taking advantage of the precision and sensitivity of ddRT-PCR technology, we show that variant-specific splicing isoforms can be expressed at very

different levels, with important consequences for interpretation of clinical significance.

The *BRCA2*:c.681+5G>C variant was previously shown to induce skipping of exon 8 (Quiles et al, 2016). Here, we found that all mRNAs expressed in homozygous cells exclude exon 8, whereas approximately half of the mRNAs expressed in heterozygous cells include exon 8. This suggests that the variant does not interfere with the expression of the WT allele. We also identified two distinct splicing

isoforms specifically transcribed from the mutant allele. In one isoform, exon 7 is spliced to exon 9 leading to a frameshift. As expected, our results are consistent with degradation of these transcripts by NMD. When degradation was prevented by a protein synthesis inhibitor, this isoform corresponded to ~29% of all *BRCA2* transcripts in heterozygous cells and ~60% of all *BRCA2* transcripts in homozygous cells. The other isoform skips most of exon 6 and exons 7 and 8 and maintains the reading frame. Accordingly, we find no evidence that these mRNAs are targeted by NMD. They represent 8–10% of all *BRCA2* transcripts in heterozygous cells and 20% of all *BRCA2* transcripts in homozygous cells.

Do the low abundant in-frame transcripts have a functional impact? To address this question, we assessed DNA repair activity in homozygous cells. We observed that compared with control cells, mutant cells formed significantly less RAD51 foci after treatment with etoposide and had more chromosome breaks induced by mitomycin C. Thus, despite the presence of potential rescue transcripts and the expression of the BRCA2 protein, cells homozygous for the *BRCA2*: c.681+5G>C variant are deficient in DNA repair activity. Consistent with this observation, the variant was clinically associated with high cancer risk, as one of the sisters developed breast cancer at the age of 29 yr, and her sibling developed colorectal cancer at the age of 38 yr. Because the *BRCA2* gene is essential for embryonic development (Suzuki et al, 1997), the patient cells cannot be completely devoid of the BRCA2 function. Moreover, the two patients did not exhibit typical features of Fanconi anemia such as congenital anomalies, bone marrow failure, and childhood cancers, and their hypersensitivity to chromosomal breakage induced by die-poxybutane (DEB) was below the threshold required for Fanconi anemia diagnosis. This suggests that the c.681+5G>C variant has a lower impact on BRCA2 function than other *BRCA2* deleterious mutations associated with Fanconi anemia.

We envisaged two possibilities to explain why BRCA2 function is reduced in the patient cells. Either the BRCA2 protein translated from in-frame truncated transcripts is fully functional, but it is not produced in sufficient amount, or the internally truncated protein is not fully functional. We thought the latter scenario was unlikely in light of a previous study showing that exons 6, 7, and 8 do not encode essential parts of the BRCA2 protein (Mesman et al, 2020). This conclusion was based on an assay that involves the introduction of human *BRCA2* into a mouse embryonic stem cell line. Variants were assessed for their ability to overcome cell lethality after conditional murine BRCA2 deletion and perform homology-directed DNA repair (Mesman et al, 2019).

Assuming that the BRCA2 protein translated from in-frame truncated transcripts is functional, we predicted that increasing its expression would suffice to revert the observed DNA repair deficiency. We used CRISPR-Cas9 gene editing to force the expression of an in-frame transcript lacking exons 7 and 8. Upon transfection of Cas9 and guide RNAs targeting exon 7 splice acceptor and donor, random insertions and deletions disrupted the canonical splice site sequences resulting in skipping of exon 7. Skipping of exon 7 in combination with the variant-induced skipping of exon 8 leads to a synthetic in-frame transcript. Compared with the parental patient-derived cells, the gene-edited cells expressed higher levels of the BRCA2 protein, formed significantly more RAD51 foci after exposure to etoposide, and showed less

chromosome breaks induced by mitomycin C. However, compared with control cells with WT *BRCA2*, the gene-edited cells did not fully recover. In this regard, it is important to note that in edited cells, the synthetic isoform lacking exons 7 and 8 represented ~55% of all *BRCA2* transcripts. This partial editing efficiency, a limitation of the present study, reflects the inherent complexity of CRISPR/Cas9-mediated gene editing. Future optimization efforts should prioritize refining gRNA design to enhance on-target efficiency and achieve higher penetrance of the desired edits. Although our results are consistent with the view that the level of DNA repair efficiency in *BRCA2* mutant cells depends on the abundance of in-frame mRNA isoforms expressed, future experiments are needed to determine whether further increasing the proportion of in-frame mRNAs leads to a complete recovery of DNA repair function.

In conclusion, our study shows that a systematic quantification of mRNA splicing isoform diversity in patient cells is crucial for the clinical interpretation of hypomorphic variants associated with the expression of out-of-frame transcripts. Furthermore, we provide evidence for the therapeutic potential of splicing modulation as a strategy to rescue BRCA2 function in mutant cells. Manipulation of pre-mRNA splicing by either antisense oligonucleotides or small molecules is a promising therapeutic approach for genetic diseases such as spinal muscular atrophy (Levin, 2019; Ratni et al, 2021). Therapeutic splicing modulation has also been achieved using CRISPR-Cas9 (Xu et al, 2019) and CRISPR-guided base editing (Gapinske et al, 2018; Yuan et al, 2018). Thus, splicing modulation has the potential to enable a new generation of personalized preventive therapies for individuals at high risk for hereditary cancer.

# Materials and Methods

### Ethics

Ethical approval for this study was obtained from the Institutional Ethics Committee of the Lisbon Academic Medical Center, Lisbon, Portugal (approval number: 170/18). Written informed consent was obtained from all patients and control individuals in accordance with European and National ethical regulation (law 12/2005). The data were treated confidentially according to (EU) 2016/679 of 27 April 2016 (General Data Protection Regulation), and the study complied with the tenets of the World Medical Association (WMA) Declaration of Helsinki for biomedical research.

### PBMCs and cell immortalization

PBMCs were isolated from fresh whole blood (collected in EDTA tubes) using the standard gradient separation approach with Histopaque-1077 (Sigma-Aldrich). A fraction of the cells (2–4 × $10^6$ cells/vial) was frozen in FBS with 10% (vol/vol) DMSO for the generation of LCLs, and the remaining cells were cultured at 37°C in 5% $CO_2$ (1 × $10^6$ cells/ml) in complete medium consisting of RPMI 1640 (Gibco) supplemented with 2 mM L-glutamine (Gibco) 15% FBS (Gibco), 1% (vol/vol) NEAA (Gibco), and 1% (vol/vol) penicillin–streptomycin (Gibco). Lymphocyte growth was stimulated with 1% (vol/vol) phytohemagglutinin (Gibco) and 0.1 μg/ml of TPA (Sigma-

Aldrich). Fresh medium was added at day 3 and at day 5. Just 4 h before harvesting the cells, each culture was split evenly, and one part was treated with 50 $\mu$g/ml cycloheximide (Sigma-Aldrich).

For cell immortalization into LCLs, cells were infected with EBV, which has been demonstrated to immortalize human resting B cells in vitro, giving rise to an actively proliferating B-cell population. Briefly, a vial of previously cryopreserved PBMCs was thawed and the cells were diluted in culture medium composed of RPMI 1640 with Hepes and L-glutamine (Gibco), 15% FBS (Gibco), 1% penicillin–streptomycin 10,000 U/ml (Gibco), and 2 $\mu$g/ml of cyclosporine A (Thermo Fisher Scientific). For EBV infection, 1 ml of PBMCs was transferred to a 24-well plate and 0.5 ml of filtered culture supernatant of the B95-8 cell line, producer of the Epstein–Barr virus, was added to the cells. The expansion of the cells was monitored weekly, and new culture medium was added gradually (1:3 dilutions) until the culture reached 40 ml. When the culture reached confluency, the cells were frozen in RPMI 1640 with Hepes and L-glutamine, 20% FBS, and 10% DMSO (1–10 × $10^6$ cells/vial), and stored in liquid nitrogen. All generated LCLs are available upon request.

### Cell culture

Control LCLs were GM12878, GM22129, GM20172, and GM16113 (Coriell Institute for Medical Research). LCLs were maintained in suspension culture, in complete medium consisting of RPMI 1640 (Gibco) supplemented with 2 mM L-glutamine (Gibco) 15% FBS (Gibco), and 1% (vol/vol) NEAA (non-essential amino acids, Gibco), at 37°C in 5% CO2 atmosphere. Before each assay, cells were cultured in fresh medium at a density of 0.6 × $10^6$ cells/ml, in 12-ml/T25 flask in a vertical position, for 20–24 h. The treatment with 50 $\mu$g/ml cycloheximide (Sigma-Aldrich) was done for 4 h before harvesting the cells. qPCR Mycoplasma Test (Mycoplasmacheck; Eurofins Genomic) was used for detection of mycoplasma in cell lines, following the manufacturer's instructions.

### Nucleus isolation

For nucleus isolation, 5 × $10^6$ cells were harvested by centrifugation at 120$g$ for 8 min and washed with ice-cold PBS. Then, cells were resuspended in 0.5 ml of ice-cold hypotonic buffer (20 mM Tris–HCl, pH 7.4; 10 mM NaCl; 3 mM $MgCl_2$) and incubated on ice for 15 min, after which 10% Nonidet P-40 was added to a final concentration of 0.5% with a vigorous mix by vortex, to lyse cell membranes. The nuclear fraction was pelleted by centrifugation for 10 min at 900$g$ and then lysed with 0.5 ml of NZYol for RNA purification.

### RNA purification

For RNA isolation, 1–2 × $10^6$ cells were washed with PBS and lysed with 0.5 ml of NZYol (MB18501; NZYTech). After RNA purification according to the manufacturer's instructions and dissolution in 10 $\mu$l of sterile RNase-free water, RNA was treated with 0.5 U/$\mu$l DNase I (Roche) in the presence of 1 U/$\mu$l RiboSafe (Bioline), an RNase inhibitor, for 30 min at 37°C, followed by further purification with acid phenol–chloroform–isoamyl alcohol (25:24:1) extraction and ethanol precipitation. Finally, RNA was dissolved

in 30 $\mu$l of RNase-free water, quantified on NanoDrop Spectrophotometer (Thermo Fisher Scientific NanoDrop 2000), and kept at –70°C.

### ddRT–PCR

Splicing isoforms were studied using One-Step RT–ddPCR Advanced Kit for Probes (reference 1864022; Bio-Rad) with 30 ng of RNA per reaction and 0.22 $\mu$M forward and reverse primers from the *BRCA2*_e5e9 set or, alternatively, from the *BRCA2*_e5e10 set (see Table 1). After partitioning the mix into aqueous droplets in oil in QX200 Droplet Generator (Bio-Rad), the following retrotranscription/amplification conditions were used: 50°C, 60 min; 95°C, 10 min; and (95°C, 35 s; 60°C, 1 min; 72°C, 4 min) x 40 cycles, then kept at 14°C. The amplification products were recovered from oil droplets according to the manufacturer's instructions, briefly: allowing the droplets to float on top, the bottom oil phase was carefully removed; then, the droplets in each reaction tube were mixed with 20 $\mu$l of TE buffer and 70 $\mu$l of chloroform and vortexed at maximum speed for 1 min, followed by 15,000$g$ centrifugation for 10 min, for phase separation, and upper aqueous phase was recovered containing amplification products. After agarose gel electrophoresis, isolated cDNA fragments were sequenced using the forward and the reverse RT–PCR primer. Briefly, size-selected fragments, extracted from agarose gel, were purified using NZYGelpure columns (NZYTech), according to the manufacturer's instructions, and eluted in 30 $\mu$l of elution buffer, and from this, 10 $\mu$l of purified sample was added to 3 $\mu$l of the corresponding 10 $\mu$M primer and sent to sequence (STAB VIDA). Sequencing results were analyzed on Chromas Lite software (Technelysium) and compared with transcript sequences described in Ensembl (hg38; Ensembl Genome Browser), using the CLUSTALW online tool. BLAST analysis was performed using the online tool (Ensemble Genome Browser).

### Quantification of splicing isoforms

Ratios of splicing isoforms were quantified using ddRT–PCR with boundary junction-spanning TaqMan fluorescent probe sets. The sets composed of one primer pair and one fluorescence-quenched TaqMan probe (5' nuclease probe) were designed with the help of IDT PrimerQuest Tool (IDT, Integrated DNA Technologies) and purchased from IDT (PrimeTime qPCR Probe Assays; ratio 3.6 (9 nmole primers: 2.5 nmole probe)). Probes are oligonucleotides containing a fluorophore at 5' end, an internal ZEN quencher at 9 nt distance, and an Iowa Black FQ quencher at 3' end, providing that double-quenching is used to reduce background signals. For each assay, a green fluorescent primer/probe set, with fluorophore being FAM (6-carboxyfluorescein or fluorescein amidite), for the inquired RNA region, and a red fluorescent primer/probe set, with fluorophore HEX (hexachloro-fluorescein), for the reference RNA region, were used. Reference control was the transcript of GUSB (glucuronidase beta), in the exon 11/exon 12 junction region.

For transcript-level quantification, QX200 Droplet Digital PCR System (Bio-Rad) and One-Step RT–ddPCR Advanced Kit for Probes (Bio-Rad) were used, according to the manufacturer's instructions

**Table 1. List of oligos for RT–PCR.**

| Target locus | Primer name | Primer/probe | DNA oligo sequence 5′-3′ | Modification | Product size (bp) |
|---|---|---|---|---|---|
| BRCA2_e5e9 | BRCA2_E5F | forward | GTACACATGTAACACCACAAAG | | 312 |
| | BRCA2_E9R | reverse | CTGTCACAGAAGCGATAAATCT | | |
| BRCA2_e5e10 | BRCA2_E5F | forward | GTACACATGTAACACCACAAAG | | 684 |
| | BRCA2_E10R | reverse | AGGGCTTCTGATTTGCTACAT | | |
| GUSB_e11e12 | GUSB_E11F | forward | GCCGATTTCATGACTGAACAG | | 80 |
| | GUSB_E12R | reverse | TTTGGTTGTCTCTGCCGAG | | |
| | GUSB_E12_PBR | probe | TCCCCTTTTTATTCCCCAGCACTCTC | HEX/ZEN/IBFQ | |
| BRCA2_e7e8 | BRCA2_E7F | forward | CTTGGTCAAGTTCTTTAGCTACAC | | 90 |
| | BRCA2_E8R | reverse | GAGGAAATACAGTTTCAGATGCTTC | | |
| | BRCA2_E7E8_PBF | probe | ACCCACCCTTAGTTCTACTGTGCTCA | 6-FAM/ZEN/IBFQ | |
| BRCA2_Δ8 | BRCA2_E7F2 | forward | GCTGAGGTGGATCCTGATA | | 149 |
| | BRCA2_E9R | reverse | CTGTCACAGAAGCGATAAATCT | | |
| | BRCA2_E7E9_PBR | probe | TCACATTCTATGAGCACAGTAGAACTAAGG | 6-FAM/ZEN/IBFQ | |
| BRCA2_Δ6q,7 | BRCA2_E5E6E8F | forward | AGAGATAAGTCAGTGTCAGAA | | 74 |
| | BRCA2_E8E9R | reverse | TTTCACATTAGCAGTAGTATCA | | |
| | BRCA2_E8_PF | probe | TGAAGAAGCATCTGAAACTGTATTTCCTCA | 6-FAM/ZEN/IBFQ | |
| BRCA2_Δ6q-8 | BRCA2_E5E6E8F2 | forward | AGAGATAAGTCAGTGAATGTGAAA | | 119 |
| | BRCA2_E9R2 | reverse | GCAGCTTCTCTTTGATTTGTG | | |
| | BRCA2_E9_P | probe | AGATTTATCGCTTCTGTGACAGACAGTGA | 6-FAM/ZEN/IBFQ | |
| BRCA2_Δ7,8 | BRCA2_E6E9F | forward | ACACCAAAGTTTGTGAAGAATG | | 184 |
| | BRCA2_E10R | reverse | TGTGGTCTTTGCAGCTATT | | |
| | BRCA2_E9E10_PBF | probe | AGAGAAGCTGCAAGTCATGGATTTGGA | 6-FAM/ZEN/IBFQ | |
| BRCA2_e13e14 | BRCA2_E13F | forward | TTCTTTAGAGCCGATTACCTGTG | | 84 |
| | BRCA2_E14R | reverse | CCAGGTGCGGTAAAATTTGG | | |
| | BRCA2_E13E14_PBR | probe | ACGTTCCTTAGTTGTGCGAAAGGGT | 6-FAM/ZEN/IBFQ | |
| BRCA2_e18e19 | BRCA2_E18F2 | forward | TGAAGCCCCAGAATCTCTTATG | | 89 |
| | BRCA2_E19R | reverse | GGTCAGGAAAGAATCCAAGTTTG | | |
| | BRCA2_E18E19_PBR | probe | CGAGCAGGCCGAGTACTGTTAGC | 6-FAM/ZEN/IBFQ | |
| BRCA2_e18i18 | BRCA2_E18F3 | forward | GTGGGCTCTCCTGATGC | | 90 |
| | BRCA2_I18R | reverse | GACTGATTTTTACCAAGAGTGCAA | | |
| | BRCA2_E18I18_PB | probe | ACACCTCTTGAAGCCCCAGAATCTC | 6-FAM/ZEN/IBFQ | |

and the assays were optimized according to MIQE guidelines (dMIQE Group & Huggett, 2020). For each assay, 60 ng of purified total RNA in 20 $\mu$l of final reaction, including 900 nM primers and 250 nM probe for each isoform-specific target and for the reference transcript, was partitioned into 20,000 water–oil emulsion droplets using QX200 Droplet Generator, and then, primer-specific retro-transcription was carried out inside each droplet, followed by PCR amplification and fluorescent hydrolysis (TaqMan) probe detection, on a C1000 thermal cycler (Bio-Rad) with the settings: 50°, 60:00; 95°, 10:00; (95°,0:35; 60° 1:00) × 45; 98°, 10 min; and 4°C. The extension step at 60°C was eventually extended to 1 min and 45 sec for assays in which the amplicon was larger than 100 bp. Primer/probe sets are listed in Table 1. End-point RT–PCR fluorescence results were read on QX200 Droplet Reader, and data were analyzed

using QuantaSoft Software version 1.7.4.0917 (Bio-Rad). Manual thresholds were applied to define positive fluorescent signal. Results are presented as the percentage of the number of copies of RNA molecules containing the inquired region relative to the number of copies of RNA molecules containing the reference region, in each assay. A non-template control was included for each primer/probe set. For each RNA sample, a blank control was performed with no reverse transcriptase in the mix (no RT).

### Immunoblot

For protein analysis, cells were washed in PBS and lysed in RIPA buffer (1 mM DTT, 1 mM EDTA, 1 mM EGTA, 150 mM NaCl, 1% NP-40, 0.5% SDC, 0.1% SDS, and 50 mM Tris–HCl, pH 7.5) supplemented with

protease inhibitors (cOmplete, Mini, EDTA-free Protease Inhibitor Cocktail; Roche). Lysis was performed for 30 min at 4°C under rotation in a volume approximately the double of the pellet of cells, followed by centrifugation at 20,000*g*, 4°C. The supernatant with the protein extract was transferred to a new tube, and protein concentration was determined by a BCA assay (Pierce BCA Protein Assay Kit; Thermo Fisher Scientific), according to the manufacturer's instructions, including a BSA (Pierce Bovine Serum Albumin Standard; Thermo Fisher Scientific) curve for each measurement.

For the electrophoretic separation via SDS–PAGE, samples were mixed with 2x Laemmli dye (20% glycerol; 4% SDS; 10% DTT; 0.02% bromophenol blue; 125 mM Tris–HCl, pH 6.8), 1% Benzonase (at 20 U/µl; Millipore), and 1% $MgCl_2$ (at 0.5 M) and incubated for 10 min at room temperature to allow for DNA degradation. Finally, the protein extracts were boiled for 10 min at 95°C and kept at −20°C. The samples (20 µg of protein and serial dilutions), as well as an appropriate protein standard marker (HiMark Prestained Protein Standard; Invitrogen), were loaded onto a 6% polyacrylamide gel. Gels were run for 90 min at 60–100 V, on running buffer (25 mM Tris; 0.19 M glycine; 0,1% SDS). For the wet transfer of the protein onto a nitrocellulose membrane (0.45 µm, Amersham Protran), a wet system was used, and the transfer was carried out at 100 V for 1 h, in transfer buffer (48 mM Tris; 38 mM glycine; 0.037% SDS; 20% methanol). The membrane was cut to probe with the different antibodies and, then, blocked with 5% BSA in PBS for 1 h under rotation. The primary antibody was incubated overnight at 4°C with rotation and diluted in antibody dilution solution (1% BSA; 0.05% Tween-20 in PBS). Membranes were washed three times for 10 min in PBST (0.05% Tween-20 in PBS). The secondary antibody was incubated for 1 h at room temperature under rotation and washed three times for 10 min, each in PBST, and 10 min in PBS. For chemiluminescence detection, membranes were incubated for 2 min with a solution composed of equal parts of solution 1 plus solution 2 from WesternBright Quantum HRP Substrate (Advansta). Excess of reagent was removed from the membrane. The drained membrane was placed on a new transparent plastic wrap, and images were acquired using Imager 680 (Amersham). Image analysis was on ImageLab 6.1 software (Bio-Rad). Primary antibodies used in this study were as follows: anti-BRCA2 (A303-434A; Bethyl Laboratories) rabbit antibody (Fortis) diluted 1:6,000; anti-CCNA2 (monoclonal EPRR19346-64) rabbit antibody (ab211735; Abcam) diluted 1:1,000; and anti-lamin A/C (E1)–HRP conjugate mouse antibody (sc-376248 HRP; Santa Cruz Biotechnology) diluted 1:2,000. Goat anti-rabbit IgG (H + L)–HRP conjugate (1706515; Bio-Rad) diluted 1:1,500 was used as a secondary antibody.

### RAD51 focus formation assay

For the functional assay, etoposide (Sigma-Aldrich), for a final concentration of 5–10 µM, was added to cells (DMSO for the mock control) with gentle resuspension and incubated for 15 min at 37°C, to induce DNA breaks. Cells were centrifuged for 5 min at 200*g*, the supernatant was removed, and then, cells were washed with warmed medium, centrifuged again, and resuspended at a density of 1 × 10^6 cells/ml. Cells were, then, distributed in 24-well plates for suspension cells, 1 ml/well, and incubated at 37°C to allow recovery

and repair pathway initiation. For immunofluorescence microscopy analysis, 10 × 10 mm² glass coverslips were previously coated with 1 mg/ml poly-L-lysine (Sigma-Aldrich). After recovery time, the plate with cells was placed on ice for 5 min to stop cell metabolism, cells were resuspended with a micropipette, and a drop with 0.1 ml (0.1 × 10^6 cells) was distributed over each coverslip and left for 10 min for cell adherence.

Afterward, cells were fixed with ice-cold 3.7% PFA in PBS for 10 min, and washed twice with PBS/0,05% Tween-20 (PBST) and then with ice-cold 70% ethanol for 10 min at room temperature. If necessary, fixed cells could be kept in ethanol solution for several weeks at −20°C. Fixed cells were washed twice with PBST, permeabilized with 0.5% Triton X-100 in PBS for 10 min with gentle swirl, washed twice with PBST and once with PBS, and blocked with blocking solution (1% BSA and 0.2% gelatin in PBST) for 20 min. The primary antibody was incubated overnight at 4°C, in a humid chamber, then washed three times for 5 min each with PBST and once with PBS, followed by incubation with secondary antibody and DAPI, for 1 h at room temperature. After three washes, for 5 min each, with PBST and once with PBS, coverslips with cells were mounted on glass slides with the antifade mounting medium, VECTASHIELD Vibrance (VectorLab). Microscopy analysis was performed on 3i Spinning Disk Confocal Microscope (Intelligent Imaging Innovations) equipped with a 63x Plan-Apochromat oil immersion objective (NA 1.40). Briefly, 20–30 z-stack images were obtained from random screen of each coverslip. Z-stacks consisted in 19–25 serial images separated by 0.27 µm, determined as optimal step size by the image software, SlideBook 6 (Intelligent Imaging Innovations). Cells with 10 or more RAD51 foci were scored as positive. At least 200 cells per experiment were scored. Quantification was performed by an experimenter who was blinded to the cell line genotype but not to treatment. Primary antibodies used were as follows: anti-phospho-histone H2AX (Ser 139) (JBW301) mouse antibody (05-636; EMD Millipore) diluted 1:500; and anti-RAD51 (monoclonal [EPR4030(3)]) rabbit antibody (ab133534; Abcam) diluted 1:250. Secondary antibodies used were as follows: horse anti-mouse IgG antibody (H+L), and DyLight 488 (DI-2488; VectorLab) and DyLight 594 anti-rabbit made in horse (DI-2489; VectorLab), both diluted 1:250. DAPI (4′,6′-diamidino-2-phenylindole; Sigma-Aldrich) is a DNA fluorescent stain used at a final concentration of 100 ng/ml.

### Chromosomal breakage analysis

Chromosomal breakage analysis was performed according to the described protocol (Oostra et al, 2012). Cells were treated with mitomycin C, at a final concentration of 150 or 300 nM, for 46 h, the last 2 h in the presence of 0.1 µg/ml colcemid to enrich in metaphases. Harvested cells, metaphase spread preparation, and Giemsa stain followed standard cytogenetic methods. Briefly, cells were pelleted by centrifugation at 200*g*, 5 min, resuspended in warm hypotonic solution (0.075 M KCl), and incubated for 20 min at 37°C. Three drops of ice-cold fixative solution (methanol: acetic acid 3:1), freshly prepared, were added before pelleting the swollen cells. Cells were resuspended in cold fixative, fixed for 20 min, and washed several times before spreading metaphases in glass slides that were, later, dried 16–72 h in the 37°C incubator. Slides were

**Table 2.  List of oligo targets for CRISPR/Cas9-mediated gene edition.**

| Target locus | gRNA name | Direction | Target sequence 5′-3′ | PAM | On-target score | Off-target score |
|---|---|---|---|---|---|---|
| BRCA2_e7p | CRISP 3 | reverse | AGACTTTCAGAAATATGTTT | TGG | 62 | 0 |
| BRCA2_e7p | CRISP 7 | reverse | GTTTTGGTGTCTGACGACCC | TGG | 70 | 77 |
| BRCA2_e7p | CRISP 1 | forward | AACTATTTTCTTTCCTCCCA | GGG | 75 | 0 |
| BRCA2_e7q | CRISP 9 | forward | CTTAGTTCTACTGTGCTCAT | AGG | 51 | 44 |

RNA oligo chemistry was Alt-R CRISPR-Cas9 crRNA.

stained with Leishman dye for 4 min, and washed with water and, then, with phosphate buffer, pH 6,8, for 2 min, followed by rapid drying at 56oC and mounting with Entellan Mounting Medium and glass coverslips. Microscopic analysis was performed by bright-field microscopy in Axiovert 200M (Zeiss) and MetaMorph software (Molecular Devices). At least 20–30 metaphases with 46 chromosomes were analyzed for each sample replicate. Chromosomal aberrations were analyzed and classified as described previously (Oostra et al, 2012). Analysis was performed by an experimenter who was blinded to the cell line genotype.

### Genomic edition mediated by CRISPR-Cas9

To achieve an efficient interference with splicing, exon 7 boundaries, on the BRCA2 gene, were targeted by guide RNA–directed CRISPR-Cas9 cleavage, driving site-specific genomic alterations because of imprecise DNA repair.

Four different RNA oligo sequences were chosen, as guides, using the IDT's Alt-R Custom CRISPR-Cas9 crRNA Design Tool and purchased from IDT (Alt-R CRISPR-Cas9 crRNA). As a criterion, the hypothesis with the highest "on-target" and "off-target" scores was selected (Table 2). The Alt-R CRISPR-Cas9 system (Integrated DNA Technologies, IDT) for genome editing and the Neon Transfection system (Invitrogen) were used following the manufacturer's instructions. Briefly, equal amounts of one specific crRNA and of the fluorescently labeled tracrRNA (IDT, Alt-R CRISPR-Cas9 tracrRNA, ATTO 550) were combined to generate one of the guide RNA complexes, followed by heating at 95°C for 5 min and cooling at room temperature for efficient hybridization; mixing the guide RNA complexes with the Alt-R Cas9 enzyme (IDT, Alt-R S.p. Cas9 nuclease), for 15 min incubation at room temperature, allowed formation of the RNP complex. Transfection of 24/20 pmol of RNP complex (RNA/Cas9, respectively) and 20 pmol of HDR Enhancer-V2 (IDT) was performed by electroporation using the Neon system (Thermo Fisher Scientific) and the Neon Transfection 10 $\mu$l Kit (Thermo Fisher Scientific), on $0.2 \times 10^6$ cells (1,550 V, 10 ms, 4 pulses). Cells recovered from transfection in conditioned medium for 24–48 h. An equal number of non-electroporated cells, whether or not in the presence of the transfection mix, were the negative controls. Flow cytometry analysis was used, 24 h after transfection, to confirm its efficiency, and, with the LSRFortessa system (BD Biosciences), to detect the entry of the fluorescently labeled ATTO 550-tracrRNA oligo into the cells.

### Selection of subclones from genome-edited cells

Monoclonal populations of edited cells were obtained by limiting dilution on two 96-well plates with a V-format and feeding with conditioned medium every 3 d. For preparation of conditioned medium, which contains cell line–specific growth factors, lymphoblastoid cells were cultured to the exponential phase of growth and pelleted by centrifugation, and the supernatant growth medium was in addition cleared from cells by filtering with a 45- $\mu$m filter (Millex-HP; Millipore). Cells growing in the wells with the highest dilutions were subsequently amplified and frozen as a stock or used in functional tests.

### Data analysis

Data analysis was performed with EXCEL (Microsoft Office); graphs and statistical analysis were performed with PRISM (GraphPad). Comparisons were performed with unpaired $t$ test analysis, not assuming consistent SD and not correcting for multiple comparisons. A $P$-value less than 0.05 was considered statistically significant. Data and statistical analysis is available as supplemental material.

## Supplementary Information

## Acknowledgements

We express our gratitude to the persons who donated blood samples for this study. We are also grateful to the Spanish National DNA Bank Carlos III of the University of Salamanca (biobank ID B.0000716 and ISCII-EU grant number PT23/00086), integrated in the "ISCIII Platform Biobanks and Biomodels," for establishing the lymphoblastoid cell lines used in this study and Genomed SA for help in PBMC purification and chromosome studies, namely, Catarina Bastos, Sónia Santos, and Sónia Matos. We also thank Carmo Cunha, Nuno Alfaiate, Sara Mijailovic, Laura Grzegorzek, Marta Sousa Santos, and Ana Margarida Veloso for helpful collaboration. We further thank the Flow Cytometry, Bioimaging, and Biobank Facilities of Instituto de Medicina Molecular João Lobo Antunes (Lisboa, Portugal) for technical support. This work was supported by Fundação para a Ciência e a Tecnologia (FCT), Portugal (PTDC/MED-ONC/3921/2021).

## Author Contributions

BA Lima: conceptualization, data curation, formal analysis, investigation, visualization, methodology, and writing—original draft.
AC Pais: investigation and methodology.
J Dupont: resources, investigation, and writing—original draft.
P Dias: resources, investigation, and writing—original draft.
N Custódio: resources, investigation, and methodology.
AB Sousa: resources, supervision, and investigation.
M Carmo-Fonseca: conceptualization, supervision, funding acquisition, project administration, and writing—original draft, review, and editing.
C Carvalho: conceptualization, data curation, formal analysis, validation, investigation, methodology, and writing—original draft, review, and editing.

## Conflict of Interest Statement

The authors declare that they have no conflict of interest.

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
