## [Reviewer comments · Life Science Alliance]

Life Science Alliance

Genetic modulation of RNA splicing rescues BRCA2 function in mutant cells

Beatriz Lima, Ana Pais, Juliette Dupont, Patrícia Dias, Noélia Custódio, Ana Sousa, Maria Carmo-Fonseca, and Celia Carvalho
DOI: <https://doi.org/10.26508/lsa.202402845>

Corresponding author(s): Celia Carvalho, Gulbenkian Institute for Molecular Medicine and Maria Carmo-Fonseca, Faculdade de Medicina da Universidade and GIMM - Gulbenkian Institute for Molecular Medicine

Review Timeline:

Submission Date:	2024-05-27
Editorial Decision:	2024-07-03
Revision Received:	2024-12-16
Editorial Decision:	2024-12-16
Revision Received:	2024-12-19
Accepted:	2024-12-20

Transaction Report:

July 3, 2024

Re: Life Science Alliance manuscript #LSA-2024-02845-T

Celia Carvalho
Instituto de Medicina Molecular João Lobo Antunes, Faculdade de Medicina da Universidade de Lisboa
Av. Prof. Egas Moniz
Lisboa, Lisboa 1649-028
PORTUGAL

Dear Dr. Carvalho,

Thank you for submitting your manuscript entitled "Genetic modulation of RNA splicing rescues BRCA2 function in mutant cells" to Life Science Alliance. The manuscript was assessed by expert reviewers, whose comments are appended to this letter. We invite you to submit a revised manuscript addressing the Reviewer comments.

Thank you for this interesting contribution to Life Science Alliance. We are looking forward to receiving your revised manuscript.

Sincerely,

B. MANUSCRIPT ORGANIZATION AND FORMATTING:

Reviewer #1 (Comments to the Authors (Required)):

In this study, the authors have examined the impact of the BRCA2 c.681+5G>C variant on splicing and BRCA2 function. This variant was previously reported to result in the skipping of exon 8, which is predicted to generate a premature termination codon in exon 9. However, the variant is classified as a variant of unknown significance (VUS) in ClinVar. Using droplet digital RT-PCR and probes from multiple BRCA2 exons, the authors show that in addition to the previously reported major transcript, a minor alternatively spliced transcript that restores the open reading frame is also expressed. This splice variant skips all but two nucleotides of exon 6 and all of exons 7 and 8. Cells from individuals who are homozygous for this variant showed reduced RAD51 foci when treated with etoposide. Similarly, these cells exhibited an increase in genomic instability after mitomycin C treatment. The authors concluded that while the new transcript restored the open reading frame, cells expressing this transcript are not proficient in BRCA2-mediated DNA repair. Based on the hypothesis that this may be due to low expression of the transcript, the authors used CRISPR/Cas9 technology to disrupt exon 7 splice acceptor and donor sites in cells homozygous for the c.681+5G>C variant. Multiple independently generated targeted cells showed expression of a splice variant lacking exons 7 and 8. This transcript represented more than half of all BRCA2 transcripts. Functional analysis revealed that the cells formed significantly more RAD51 foci in response to etoposide treatment and showed a reduction in mitomycin C-induced chromosomal aberrations compared to the parental cells. However, the rescue of the aberrant phenotype was less when compared to cells expressing full-length BRCA2.

The findings are significant and relevant to understanding the functional significance of BRCA2 VUSs. Also, CRISPR-mediated restoration of the reading frame by forced skipping of additional exons known to be dispensable for BRCA2 DNA repair function. However, the failure to completely rescue the defect suggests either the mutant protein is not fully functional and these exons are not fully dispensable for protein function, or the amount of mutant protein is not sufficient. This can be addressed by comparing the protein levels compared to WT or heterozygous cells (as they have shown in Figure 4A). Does the RAD51 foci number correlate with the protein expression levels in different clones?

A few other minor points:

Line 190: The authors have written "...out-of-frame ligation of exons 7 and 9...". "Ligation" should be changed to "splicing".

Figure 5F: Why are the points for each sample connected by a line? The data can be represented more clearly by bar graphs for each clone for each break event group.

It will be helpful to mention in the figure (not just in the figure legend) that Figure 3C represents total RNA and Figure 3D represents nuclear RNA analysis.

Reviewer #2 (Comments to the Authors (Required)):

Lima et al. have utilized ddPCR to analyze primarily a BRCA2 variant c.681+5G>C [NM_000059.4] in a breast cancer patient in heterozygosity and in two siblings, one with colon cancer and one with breast cancer who are homozygous for this variant. This variant, with a single nucleotide change within an intron, leads to weakening of proper splicing, exon skipping and introduction of an out-of-frame premature stop codon (PTC) and subsequent nonsense-mediated decay (NMD) associated with loss of BRCA2 function. For the patients with the homozygous variant there is up to 50% loss of BRCA2 mRNA by NMD and loss of BRCA2 function evaluated by lack of RAD51 foci after etoposide incubation in an immortalized lymphoid cell line from the one patient with colon cancer.

The authors have effectively utilized CRISPR/Cas9 gene editing with non-homologous end-joining (NHEJ) to induce exon skipping of exon 7 within the context of the homozygous c.681+5G>C variant to elicit skipping of non-essential E7 and E8 sequences allowing for restored in-frame BRCA2 sequence and partial restoration of BRCA2 function via RAD51 foci after etoposide as well as reduction of mitomycin-induced chromosomal breaks.

Overall, the authors have shown proof-of-concept that gene-editing has potential to restore DNA damage repair function in those individuals at risk of developing cancer related to the homozygous BRCA2 variant c.681+5G>C.

The careful use of various primer/probe sets and ddPCR effectively set the stage for the gene-editing experiments and subsequent characterizations of clones with partially restored BRCA2 function.

Despite these notable strengths, there are various concerns and some fundamental issues with the nature of the c.681+5G>C variant that the authors could and should address/reconcile.

1. Lines 142-144. The c.681+5G>C is stated to "substitute a G to C five nucleotides downstream of the natural splice site at the 3' end of exon 7 (Fig. 1B)". The variant, based on NM_000059.4, is within intron 7 and is within a spliceosome recognition domain. It would be helpful to state this more clearly rather than the cryptic statement in the text.

Importantly, Fig 1B shows this variant to be within Intron 8 NOT Intron 7. The authors cite Quiles et al. 2016 for the indication that Intron 8 is skipped with a PTC within exon 9. The authors need to reconcile this nomenclature and potential error either in their indication of the variant base change within intron 7 (based on NM_000059.4) or a possible error in the Quiles et al. paper which is carried forward in the rest of this submitted manuscript. This will be important for readership and accuracy in this field moving forward.

2. Figs. 2 and 3: ddPCR results are indicated as "proportion" of various isoforms based on primer/probe sets as numerators and either e13/314 or e18/e19 or e18/I18 as denominators. In contrast, text refers to percentage expression. It would be helpful to reader to label ordinates on figures as percentages for consistency.

For nuclear RNA analyses in Fig. 3D, the e18/I18 boundary primer/probe was used as denominator. The authors should explain why this primer/probe was utilized and why this is appropriate in the context of nuclear RNAs (pre-mRNA evaluations??).

Importantly, no statistical considerations are apparent for Fig. 3 B-D. Using the $\Delta 8$ probe in Fig. 3C, it was demonstrated that ~60% of mRNAs had skipped exon 8 and undergone NMD; a seminal finding for this manuscript upon which the gene-editing experiments proceeded. Here, statistics are required.

For Fig. 2 and Suppl Fig. 1 individual data points are indicated. In Fig. 3, results are shown as bars with errors. The authors should show individual data points for Fig. 3 for consistency.

3. For Fig. 2C (line 172) the authors indicate that forward primer on e3 and reverse primer e10 were used along with sanger sequencing of isolated amplicons to identify an additional minor variant that is in-frame and might account for some BRCA2 function. For stringency, please indicate primers sets.

4. For Fig. 2D, the authors indicate that NMD for the minor variant [6p spliced to 9] occurs. However, results indicate that there is no statistically significant increase after cycloheximide. The authors should modify their statements (lines 183-184).

5. Fig. 4A clearly shows that the immortalized lymphoid line from patient CC homozygous for the variant contains BRCA2 protein equivalent to control lymphoid cells. As a suggestion, definitive indications of exon skipping could be determined if a specific antibody was raised against the amino acids that are putatively missing.

6. Fig. 4B experiments were performed at least twice according to the authors. Results show many data points from populations with greater than 10 gamma-H2AX foci. The authors should explain better where these multiple points are from (multiple fields from one-two experiments?).

There is no statistically significant difference in etoposide induced DNA damage in Control cells compared to CC patient cells 3 hr after removal of drug. At 24 hr, all gamma-H2AX signal is gone in both Control and patient cells (Suppl. Fig. 2). It would be of interest to know what happens to DNA repair (gamma-H2AX levels) at various times between 3 hr and 24 hours to correlate with BRCA2 function and results in Fig. 4C where RAD51 foci are attenuated in CC patient cells.

7. Fig. 4C clearly demonstrates that CC patient lymphoid cell lines are deficient in RAD51 foci formation consistent with lack of BRCA2 function related to the effects of the c.681+5G>C variant. In addition, this result demonstrated that minor in-frame variant with 6p ligated to 9 (Fig. 2C) does not contribute substantially to BRCA2 function. The authors should comment on the apparent complete absence of RAD51 foci despite the maximum out-of-frame exon deletion of 50-60% for the c.681+5G>C variant.

8. Beginning on Line 277, the authors state/recapitulate that, "50-60% of BRCA2 mRNAs expressed in homozygous mutant cells contain exon 7 ligated to exon 9 generating a premature stop codon, whereas ~20% of transcripts encode an internally truncated yet potentially functional protein". It would be helpful for readership to have the authors include/point to the important Figs. where these results were demonstrated (maybe Fig. 3C,3D- $\Delta 8$ and Fig. 3C,3D $\Delta 6q-8$, respectively??).

9. Lines 288-289. Where is it noted that "the combined skipping of BRCA2 exons 7 and 8 rescues the correct reading frame". Though this is logical, where is this demonstrated aside from the minor variant 6p spliced to 9? The authors should be more clear on this issue. This seems important as one rationale for proceeding to the gene-editing experiments.

10. For the CRISPR/Cas9 gene-edited clones derived containing deleted exons 7 and 8 (Clones 1,4,5,9,12,15,16), the authors indicated that sequencing confirmed the expression of this internally truncated isoform. It would be appropriate to show these

sequences in a Supplemental figure.

11. Fig. 5C demonstrates partial restoration of BRCA2 function based on RAD51 foci for Clones 1,12, and 15 but not for clones 4 and 9. This is at odds with Fig. 5F and 5G where mitomycin C-induced chromosome breakage is reduced for all clones tested including clone 4 and 9. The authors could and should discuss this issue.

12. Supplemental Fig. 3B results with $\Delta 6q-8$ and $\Delta 8$ are not addressed but should be indicated.

13. Based on Fig. 5B and use of gRNA7, ~55% of all BRCA2 transcripts contain the internally truncated isoform lacking e7 and e8. The authors should discuss potential reasons why only about 1/2 of the transcripts contain the desired internal edits. Is there off-targeting of gRNA7? How do the authors propose in future experiments to improve the penetrance of this gene-editing approach?

14. Lines 322-324. Do the authors mean that in CONTROL cells when 10 μ M etoposide is used there are more cell RAD51 foci positive compared to cells treated with 5 μ M etoposide?

Referee Cross-Comments: I am in agreement with Reviewer #1

RESPONSE TO REVIEWERS

We thank the reviewers for their very thorough review of our work and for their constructive comments and suggestions, which have substantially improved the quality of the manuscript. We have addressed each comment (*italics*) with our specific responses below. Changes to the manuscript text are highlighted in red in the revised version.

Reply to reviewer #1:

We thank the reviewer for his/her supporting comments.

the failure to completely rescue the defect suggests either the mutant protein is not fully functional and these exons are not fully dispensable for protein function, or the amount of mutant protein is not sufficient. This can be addressed by comparing the protein levels compared to WT or heterozygous cells (as they have shown in Figure 4A). Does the RAD51 foci number correlate with the protein expression levels in different clones?

R: We now include a Western blot showing increased BRCA2 protein levels in the edited clones compared to the parental patient cells (Suppl. Fig. 3D). The text was revised to include and discuss these new results.

"Line 190: The authors have written "...out-of-frame ligation of exons 7 and 9...". "Ligation" should be changed to "splicing".

R: This has been changed throughout the manuscript.

Figure 5F: Why are the points for each sample connected by a line? The data can be represented more clearly by bar graphs for each clone for each break event group.

R: We totally agree, and we have replaced this panel, now including bar graphs.

It will be helpful to mention in the figure (not just in the figure legend) that Figure 3C represents total RNA and Figure 3D represents nuclear RNA analysis.

R: We thank the reviewer for this suggestion, and we modified Fig. 3 accordingly.

Reply to reviewer #2:

We thank the reviewer for his/her supporting comments.

1. Lines 142-144. The c.681+5G>C is stated to "substitute a G to C five nucleotides downstream of the natural splice site at the 3' end of exon 7 (Fig. 1B)". The variant, based on NM_000059.4, is within intron 7 and is within a spliceosome recognition domain. It would be helpful to state this more clearly rather than the cryptic statement in the text.

Importantly, Fig 1B shows this variant to be within Intron 8 NOT Intron 7. The authors cite Quiles et al. 2016 for the indication that Intron 8 is skipped with a PTC within exon 9. The authors need to reconcile this nomenclature and potential error either in their indication of the variant base change within intron 7 (based on NM_000059.4) or a possible error in the Quiles et al. paper which is carried forward in the rest of this submitted manuscript. This will be important for readership and accuracy in this field moving forward.

R: Thank you for pointing out this oversight. We have now provided a more detailed description of the variant in the revised manuscript and have corrected the error to specify that the variant is located in intron 8, not intron 7 as previously stated. Our initial statement indicating that the variant was located in intron 7 was based on information from the Ambry Genetics and ClinVar annotations, which describe the c.681+5G>C variant as "a G to C substitution 5 nucleotides after coding exon 7 in the BRCA2 gene" (<https://www.ncbi.nlm.nih.gov/clinvar/variation/483115/>). Upon further investigation, we realized this was incorrect, as the variant is actually located in intron 8, as confirmed by the LOVD database (https://databases.lovd.nl/shared/refseq/BRCA2_NM_000059.3_table.html). We have now corrected this error in the manuscript and clarified the accurate location and nomenclature of the variant.

2.1.Figs. 2 and 3: ddPCR results are indicated as "proportion" of various isoforms based on primer/probe sets as numerators and either e13/314 or e18/e19 or e18/I18 as denominators. In contrast, text refers to percentage expression. It would be helpful to reader to label ordinates on figures as percentages for consistency.

R: Ordinates on figures were re-labelled as “percentages”, as suggested.

2.2. For nuclear RNA analyses in Fig. 3D, the e18/i18 boundary primer/probe was used as denominator. The authors should explain why this primer/probe was utilized and why this is appropriate in the context of nuclear RNAs (pre-mRNA evaluations??).

R: Correct, we used an exon/intron boundary primer to specifically detect unspliced transcripts in the nucleus. This is now explained in the figure legend.

2.3. Importantly, no statistical considerations are apparent for Fig. 3 B-D. Using the $\Delta 8$ probe in Fig. 3C, it was demonstrated that ~60% of mRNAs had skipped exon 8 and undergone NMD; a seminal finding for this manuscript upon which the gene-editing experiments proceeded. Here, statistics are required.

R: We totally agree. We performed additional experimental replicates and we now include statistics in all panels.

2.4. For Fig. 2 and Suppl Fig. 1 individual data points are indicated. In Fig. 3, results are shown as bars with errors. The authors should show individual data points for Fig. 3 for consistency.

R: New panels are provided in Fig. 3 showing individual data points.

3. For Fig. 2C (line 172) the authors indicate that forward primer on e3 and reverse primer e10 were used along with sanger sequencing of isolated amplicons to identify an additional minor variant that is in-frame and might account for some BRCA2 function. For stringency, please indicate primers sets.

R: The primer pair used for sequencing $\Delta 8$ isoform was forward and reverse on exons 5 and 9, respectively, named BRCA2 e5e9 set in Methods, and presented in Table 1. The primer pair used for sequencing $\Delta 6q-8$ isoform was forward and reverse on exons 5 and 10, respectively.

This information is included in the Methods section, Table 1, and Fig. 2C.

4. For Fig. 2D, the authors indicate that NMD for the minor variant [6p spliced to 9] occurs. However, results indicate that there is no statistically significant increase after cycloheximide. The authors should modify their statements (lines 183-184).

R: We thank the reviewer for raising this question. We performed additional replicates, and the difference is now statistically significant.

5. Fig. 4A clearly shows that the immortalized lymphoid line from patient CC homozygous for the variant contains BRCA2 protein equivalent to control lymphoid cells. As a suggestion, definitive indications of exon skipping could be determined if a specific antibody was raised against the amino acids that are putatively missing.

R: We appreciate this excellent suggestion. However, despite our efforts, we have not yet been able to generate or obtain a specific antibody. Developing such an antibody remains a priority for future work.

6.1. Fig. 4B experiments were performed at least twice according to the authors. Results show many data points from populations with greater than 10 gamma-H2AX foci. The authors should explain better where these multiple points are from (multiple fields from one-two experiments?).

R: We now explain in the Figure legends that “Dot plots depict results from multiple assays, with each dot representing an individual replicate.”

6.2. There is no statistically significant difference in etoposide induced DNA damage in Control cells compared to CC patient cells 3 hr after removal of drug. At 24 hr, all gamma-H2AX signal is gone in both Control and patient cells (Suppl. Fig. 2). It would be of interest to know what happens to DNA repair (gamma-H2AX levels) at various times between 3 hr and 24 hours to correlate with BRCA2 function and results in Fig. 4C where RAD51 foci are attenuated in CC patient cells.

R: We appreciate the reviewer's insightful suggestion to investigate DNA repair dynamics (γ -H2AX levels) at various time points between 3 and 24 hours. We agree that this may provide valuable additional information, and we consider it an interesting avenue for future research.

7. Fig. 4C clearly demonstrates that CC patient lymphoid cell lines are deficient in RAD51 foci formation consistent with lack of BRCA2 function related to the effects of the c.681+5G>C variant. In addition, this result demonstrated that minor in-frame variant with 6p ligated to 9 (Fig. 2C) does not contribute substantially to BRCA2 function. The authors should comment on the apparent complete absence of RAD51 foci despite the maximum out-of-frame exon deletion of 50-60% for the c.681+5G>C variant.

R: We totally agree. In the Discussion section, we comment "despite the presence of potential rescue transcripts and expression of BRCA2 protein, cells homozygous for the BRCA2:c.681+5G>C variant are deficient in DNA repair activity." (lines 391-392)

8. Beginning on Line 277, the authors state/recapitulate that, "50-60% of BRCA2 mRNAs expressed in homozygous mutant cells contain exon 7 ligated to exon 9 generating a premature stop codon, whereas ~20% of transcripts encode an internally truncated yet potentially functional protein". It would be helpful for readership to have the authors include/point to the important Figs. where these results were demonstrated (maybe Fig. 3C,3D-Δ8 and Fig. 3C,3D Δ6q-8, respectively??).

R: We thank the reviewer for this suggestion. We have included reference to the Figures in the text.

9. Lines 288-289. Where is it noted that "the combined skipping of BRCA2 exons 7 and 8 rescues the correct reading frame". Though this is logical, where is this demonstrated aside from the minor variant 6p spliced to 9? The authors should be more clear on this issue. This seems important as one rationale for proceeding to the gene-editing experiments.

R: To clarify this point, we have revised the manuscript to explicitly state that “Upon analyzing the BRCA2 gene sequence, we noted that the simultaneous skipping of exons 7 and 8 preserves the reading frame.” (lines 291-292)

10. For the CRISPR/Cas9 gene-edited clones derived containing deleted exons 7 and 8 (Clones 1,4,5,9,12,15,16), the authors indicated that sequencing confirmed the expression of this internally truncated isoform. It would be appropriate to show these sequences in a Supplemental figure.

R: Sequencing results were added to supplementary Figure 3C.

11. Fig. 5C demonstrates partial restoration of BRCA2 function based on RAD51 foci for Clones 1,12, and 15 but not for clones 4 and 9. This is at odds with Fig. 5F and 5G where mitomycin C-induced chromosome breakage is reduced for all clones tested including clone 4 and 9. The authors could and should discuss this issue.

R: We have added a discussion of this discrepancy to the manuscript (lines 341-354).

12. Supplemental Fig. 3B results with $\Delta 6q-8$ and $\Delta 8$ are not addressed but should be indicated.

R: We now address these results (lines 316-318).

13. Based on Fig. 5B and use of gRNA7, ~55% of all BRCA2 transcripts contain the internally truncated isoform lacking e7 and e8. The authors should discuss potential reasons why only about 1/2 of the transcripts contain the desired internal edits. Is there off-targeting of gRNA7? How do the authors propose in future experiments to improve the penetrance of this gene-editing approach?

R: We thank the reviewer for this insightful comment. We have added this discussion to the manuscript (lines 423-427).

14. Lines 322-324. Do the authors mean that in CONTROL cells when 10 μ M etoposide is used there are more cell RAD51 foci positive compared to cells treated with 5 μ M etoposide?

R: The reviewer is correct. We have clarified this point in the revised text (lines 328-331) to explicitly state that treatment with 10 μ M etoposide results in a higher proportion of RAD51 foci-positive cells compared to treatment with 5 μ M etoposide.

December 16, 2024

RE: Life Science Alliance Manuscript #LSA-2024-02845-TR

Prof. Celia Carvalho
Gulbenkian Institute for Molecular Medicine
Av. Prof. Egas Moniz
Lisboa, Lisboa 1649-028
Portugal

Dear Dr. Carvalho,

Thank you for submitting your revised manuscript entitled "Genetic modulation of RNA splicing rescues BRCA2 function in mutant cells". We would be happy to publish your paper in Life Science Alliance pending final revisions necessary to meet our formatting guidelines.

- please be sure that the authorship listing and order is correct
- please add the Twitter handle of your host institute/organization as well as your own or/and one of the authors in our system
- please be sure that the authorship listing and order is correct
- please add ORCID ID for the secondary corresponding author--they should have received instructions on how to do so
- please use the [10 author names et al.] format in your references (i.e., limit the author names to the first 10)
- please add an Author Contributions section to your main manuscript text
- please move your supplementary figure legends in the main manuscript text after the legends for the main figures and before the tables
- please remove the section entitled "The paper explained"

FIGURE CHECKS:

- please add sizes next to the blot in Figure S3D

LSA now encourages authors to provide a 30-60 second video where the study is briefly explained. We will use these videos on social media to promote the published paper and the presenting author (for examples, see <https://docs.google.com/document/d/1-UWCfbE4pGcDdcgzcmiuJl2XMBJnxKYeqRvLLrLS08s/edit?usp=sharing>). Corresponding or first-authors are welcome to submit the video. Please submit only one video per manuscript. The video can be emailed to contact@life-science-alliance.org

A. FINAL FILES:

B. MANUSCRIPT ORGANIZATION AND FORMATTING:

Sincerely,

December 20, 2024

RE: Life Science Alliance Manuscript #LSA-2024-02845-TRR

Prof. Celia Carvalho
Gulbenkian Institute for Molecular Medicine
Av. Prof. Egas Moniz
Lisboa, Lisboa 1649-028
Portugal

Dear Dr. Carvalho,

Thank you for submitting your Research Article entitled "Genetic modulation of RNA splicing rescues BRCA2 function in mutant cells". It is a pleasure to let you know that your manuscript is now accepted for publication in Life Science Alliance. Congratulations on this interesting work.

DISTRIBUTION OF MATERIALS:

Again, congratulations on a very nice paper. I hope you found the review process to be constructive and are pleased with how the manuscript was handled editorially. We look forward to future exciting submissions from your lab.

Sincerely,
